# Characterization of the Complete Mitochondrial Genome of the Spotted Catfish *Arius maculatus* (Thunberg, 1792) and Its Phylogenetic Implications

**DOI:** 10.3390/genes13112128

**Published:** 2022-11-16

**Authors:** Min Yang, Zimin Yang, Cuiyu Liu, Xuezhu Lee, Kecheng Zhu

**Affiliations:** 1Joint Laboratory of Guangdong Province and Hong Kong Region on Marine Bioresource Conservation and Exploitation, College of Marine Sciences, South China Agricultural University, Guangzhou 510642, China; 2Key Laboratory of South China Sea Fishery Resources Exploitation and Utilization, Ministry of Agriculture, South China Sea Fisheries Research Institute, Chinese Academy of Fishery Sciences, Guangzhou 510300, China

**Keywords:** *Arius maculatus*, mitochondrial genome, RSCU, Ka/Ks, phylogenetic analysis

## Abstract

The spotted catfish, *Arius maculatus* (Siluriformes), is an important economical aquaculture species inhabiting the Indian Ocean, as well as the western Pacific Ocean. The bioinformatics data in previous studies about the phylogenetic reconstruction of Siluriformes were insufficient and incomplete. In the present study, we presented a newly sequenced *A. maculatus* mitochondrial genome (mtDNA). The *A. maculatus* mtDNA was 16,710 bp in length and contained two ribosomal RNA (rRNA) genes, thirteen protein-coding genes (PCGs), twenty-two transfer RNA (tRNA) genes, and one D-loop region. The composition and order of these above genes were similar to those found in most other vertebrates. The relative synonymous codon usage (RSCU) of the 13 PCGs in *A. maculatus* mtDNA was consistent with that of PCGs in other published Siluriformes mtDNA. Furthermore, the average non-synonymous/synonymous mutation ratio (Ka/Ks) analysis, based on the 13 PCGs of the four Ariidae species, showed a strong purifying selection. Additionally, phylogenetic analysis, according to 13 concatenated PCG nucleotide and amino acid datasets, showed that *A. maculatus* and *Netuma thalassina* (Netuma), *Occidentarius platypogon* (Occidentarius), and *Bagre panamensis* (Bagre) were clustered as sister clade. The complete mtDNA of *A. maculatus* provides a helpful dataset for research on the population structure and genetic diversity of Ariidae species.

## 1. Introduction

In metazoans, the typical complete mitochondrial genome (mtDNA) is usually a circular, double-stranded molecule, with sizes ranging from 13 to 20 kb, containing 13 protein-coding genes (PCGs), 2 ribosomal RNA (rRNA) genes, and 22 transfer RNA (tRNA) genes. Moreover, an A + T-rich region, which includes the initial sites for RNA transcription and mtDNA replication, is regarded as the non-coding region or the control region (CR) [1]. Owing to its maternal inheritance, rapid evolutionary rate, short coalescence time, conserved gene content, small genome size, and low levels of sequence recombination [2,3], mtDNA is widely used in various research fields, such as species identification and taxonomic resolution [4,5], comparative and molecular evolution [6,7,8], population genetics [9], and non-synonymous (Ka) and synonymous ㉿ substitutions of many species [5,10,11,12].

Moreover, mtDNA is commonly known as a helpful molecular marker for phylogenetic analyses among fish taxa. A single mitochondrial gene fragment has limitations in resolving complex phylogenetic relationships in plentiful fish lineages [13]. The additional informative sites from complete mtDNA allow these detailed branches and higher-level relationships to be more adequately resolved [14]. Consequently, in the present study, the mtDNA data may improve the understanding of the evolutionary relationships of the family Siluriformes.

*Arius maculatus*, also known as the spotted catfish, belongs to Ariidae, Siluriformes. It is a benthic species in subtropical and tropical waters, inhabiting the bottoms of rivers, estuaries, and coasts, and is extensively distributed in the Indo-West Pacific (http://fishbase.sinica.edu.tw/Summary/SpeciesSummary.php, accessed on 4 March 2019) [15]. It has a long body shape, broad front, lateral flat rear, and a body length of more than 60 cm. Furthermore, there are serrated poison glands at the base of the dorsal and pectoral spines, which cause severe pain when stabbed, and are the defensive tools of the fish. The fish has a strong smell, but it has a high fat content. Southeast coastal residents commonly cook it with “angelica” and other traditional Chinese medicine. The feeding strategy, morphology, and ecology of this important species has been studied in recent years [16].

A better understanding of Siluriformes mtDNAs requires expanded taxon sampling. Siluriformes includes approximately 3093 described species, classified into 478 genera and 36 superfamilies [17]. Ariidae includes approximately 26 genera and over 133 species [17,18], many of which are agriculturally important. The classification of Ariidae species is arguably the most poorly resolved of any catfish family [19]. Simply, the subfamilial divisions within the Ariidae (Galeichthyinae and Ariinae) were absolutely consistent among the four reconstruction methods conducted (MP, BI, ML-RAxML, and ML-Garli) and well-supported [20]. Moreover, mtDNA synteny analysis has revealed many common mtDNA features in Siluriformes, which may lead to a better understanding of the evolution of Siluriformes [21]. Despite the vast species diversity in this family, there are only five species containing available complete mtDNAs in the GenBank database, and mtDNA information in the family Ariidae is only available for three species, *Occidentarius platypogon* [22], *Bagre panamensis* [23], and *Arius arius* [24]. To date, there is still an observable lack of mtDNA information among Ariidae, and the phylogenetic relationships and taxonomic status of *A. maculatus* in Ariidae are still vague. Consequently, to understand the evolutionary relationships of *A. maculatus* in Siluriformes and further study the population genetics in Ariidae, we sequenced the complete mtDNA of *A. maculatus* and analyzed its characteristics and evolutionary relationships.

## 2. Materials and Methods

### 2.1. Samples and Mitogenome Sequencing

Adult *A. maculatus* fish (about 2400 g) was obtained from Beibu Gulf, China (longitude 21°36′50″ N and latitude 108°44′00″ E) in June 2019 and directly frozen. Genomic DNA was extracted from muscle tissues, according to the instructions of Genomic DNA Extraction Ver.5.0 kit (TaKaRa, Kyoto, Japan). The concentration of the isolated gDNA was detected using the NANODROP 2000 spectrophotometer (Thermo Scientific, Waltham, MA, USA). The quality of the extracted gDNA was evaluated by electrophoresis with 1% agarose gel and stained with Gel Red™ (Biotium). Then, normalized genomic DNA (4 μg) was used to prepare the paired-end library, according to the instructions of the NEBNext DNA sample libraries kit (NEB, New England). The quantification and size of the library was estimated using a Bioanalyzer 2100 High Sensitivity DNA chip (Agilent, CA, USA). Sequencing of the normalized library (2 nM) was performed on a HiSeq 2500 platform (2 × 101 bp paired-end reads) (San Diego, IL, USA).

### 2.2. Genome Assembly and Annotation

Clean data were generated according to a previous protocol [25], and the remaining high-quality reads were then assembled using SeqMan NGen (http://www.dnastar.com/t-tutorials-seqman-ngen.aspx, accessed on 10 March 2021) (DNASTAR Inc., Madison, WI, USA). Match spacing, minimum match percentage, match size, gap penalty, mismatch penalty, expected genome length, and maximum gap length were set to 10, 93, 50, 30, 20, 16,000, and 6%, respectively. After alignment with the NCBI nt database, the sequences were aligned using the blastn method (https://blast.ncbi.nlm.nih.gov/, accessed on 10 March 2021). Siluriformes mtDNA-mapped sequences were identified as *A. maculatus* mtDNA. To demonstrate the preciseness of the assembled genome sequence, primers were used to amplify mtDNA (Appendix A). PCR amplification has been previously described [4]. Moreover, PCGs, rRNA genes, tRNA genes, and the D-loop region of mtDNA were annotated by the Mito Annotator (http://mitofish.aori.u-tokyo.ac.jp/annotation/input.html, accessed on 8 May 2021), according to circular genome parameters [26].

### 2.3. Genome Sequence Analysis

To confirm tRNAs, the tRNAscan-SE Search Server 1.21 program was used [27,28]. OGDRAW1.2 was used to create the gene map of *A. maculatus* mtDNA, and hand annotation was completed [29]. An estimate of strand skew was developed using a previous study’s formulae [30]. By using “models- > Compute Codon Usage Bias” in MEGA 6.0, relative synonymous codon usage (RSCU) was calculated [31]. The nonsynonymous/synonymous mutation (Ka/Ks) ratio and codon usage in the 13 PCGs were calculated using DnaSP 5.10.01 to investigate the evolutionary branching of the Ariidae lineage [32]. In addition, we determined the skew of AT and GC in the whole mtDNA, PCG, tRNA, rRNA, or control region sequence, using the following formula: AT skew = (A − T) /(A + T) and GC skew = (G − C)/ (G + C) [33].

### 2.4. Phylogenetic Analysis

Phylogenetic analysis of Siluriformes was carried out using 13 PCG nucleotide and amino acid sequences from 21 species. Based on MUSCLE v.3.8.31, each of the 13 PCG nucleotide and amino acid sequences from all 21 species was individually aligned (http://www.drive5.com/muscle/, accessed on 8 May 2021) [34] and then aggregated into a sequence matrix to reconstruct the phylogeny. The 21 mitogenome data were all downloaded from the NCBI database. Twenty-one species were divided into 15 genera and 7 families in the order Siluriformes. To test the nucleic acid and amino acid models, we used jModelTest2.1.7 (https://code.google.com/p/jmodeltest2/, accessed on 8 May 2021) [35] and Prottest3.2 [36]. Akaike information criterion(AIC), was considered the best model for tree formation. The maximum likelihood (ML) tree was implemented in RAxML 8.0.12 [37] under the GTR-γ model and MtMam+I+G model for nucleic acid and amino acid trees, respectively, and node support was calculated with 1000 bootstrap replications (random seed value of 1,234,567). Further, MrBayes 3.2.5 [38] for Bayesian inference (BI) was used to reconstruct phylogenetic trees with 10,000,000 generations. The BI analysis used the CAT-GTR model, and two independent Markov chain Monte Carlo (MCMC) chains were run for 10,000 cycles. Phylogenetic trees were generated using FigTree v1.4.2 (http://tree.bio.ed.ac.uk/software/figtree/, accessed on 8 May 2021).

## 3. Results and Discussion

### 3.1. Genome Size and Organization

Raw data of approximately 1.5 G with read lengths of 150 bp were generated. The mtDNA sequences covered 100% of the genome and were approximately 57X deep. The total number of bases (bp) was 965,100, and the read number was 6434. The whole mtDNA of *Arius maculatus* was a circular double-chain DNA molecule with a length of 16,710 bp (GenBank: MN604079; Figure 1, Table 1), which is comparable to the mtDNA of other Siluriformes species, ranging from 16,471 bp *(Pangasius larnaudii*) to 16,830 bp *(Ariopsis seemanni*) [39] (Appendix A). Nucleotide BLAST of the complete *A. maculatus* mtDNA against other Siluriformes mtDNA showed sequence homology of 99.80 (*N. thalassina*), 99.74 (*A. arius*), 90.31 (*A. seemanni*), and 90.14% (*O. platypogon*) with closely related species, and of 83.25 (*Glaridoglanis andersonii*), 83.18 (*Silurus soldatovi*), 83.15 (*S. meridionalis*), and 83.12% (*Silurus asotus*) with distantly related species (Appendix A). Moreover, the mtDNA of *A. maculatus* comprised 2 rRNA genes, 13 PCGs, 22 tRNA genes, and a D-loop region. The arrangement of the genes in the *A. maculatus* mtDNA was identical to that of other reported Ariidae mtDNAs (Table 1) [22,23,24]. Of these genes, 29 (12 PCG, 2 rRNA, and 15 tRNA) were located in the heavy strand (H-strand); the rest (1 PCG and 8 tRNA) were located in the light strand (L-strand) (Table 1). As valid species markers and genus authentication features, these typical features have also been observed in other Siluriformes [39,40,41,42].

The nucleotide composition of the mtDNA was A (29.63%), T (25.42%), C (29.65%), and G (15.30%), with a high A + T nucleotide content (55.05%), which was 54.88, 55.52, 52.75, and 62.55% for the PCGs, tRNAs, rRNAs, and D-loop region, respectively (Table 2). Among Siluriformes, *A. maculatus* had the lowest A + T nucleotide composition. With more As than Ts, the AT skew (0.0764) observed here was similar to that in *O. platypogon* (0.0765), *N. thalassina* (0.0775), and *B. panamensis* (0.0716), which are evolutionarily closely related. Most Siluriformes, however, exhibited a positive AT skew in their mtDNA (Table 2). GC skews ranged from −0.3308 in *O. platypogon* to −0.2799 in *S. soldatovi* (Table 2). In *A. maculatus*, the mtDNA was negative (−0.3194), showing that it had a GC skew more toward Cs than Gs.

### 3.2. Protein-Coding Gene Features

The PCG sequences make up 68.26% of the *A. maculatus* mtDNA, with 11,407 bp. Furthermore, 19 Siluriformes were shown to have AT skews and GC skews that differed from nucleotide composition (Table 2). Among Siluriformes species, *A. maculatus* mtDNA had a moderate AT skew value (0.0489) of the PCG region. Other species, however, showed a negative GC skew (−0.3888) [43,44,45]. In addition, thirteen PCGs with AT and GC skews were also calculated in Appendix A, indicating that they were mutually consistent and closely related [43,45]. 

With the exception of COXI, which starts with a GTG codon, each PCG is initiated by a classic ATN codon (Table 1). Other Ariidae fish have shown similar results. In six of the thirteen PCGs (ND1, COXI, ATP8, ATP6, ND4L, and ND5), a typical termination codon (TAA) is used, which is common to Siluriformes mtDNA [19,21]. COXII, COXIII, ND4, and Cytb, on the other hand, terminate with the incomplete termination signal T, whereas ND2, ND3, and ND6 terminate with TAG. (Table 1). The mtDNA of *A. maculatus* is homologous to sequenced mtDNAs of other Siluriformes, including *A. arius* [24], *O. platypogon* [24], *Pseudecheneis immaculatus* [46], and *Ailia coila* [47].

*A. maculatus* encodes 3792 amino acids in its 13 PCGs. Moreover, codon usage is displayed in Table 3. *A. maculatus* PCGs were dominated by the following amino acids: leucine (Leu, 17.1%), alanine (Ala, 8.84%), isoleucine (Ile, 8.07%), and threonine (Thr, 7.68%), whereas those encoding cysteine (Cys, 0.71%) were rare (Table 3). RSCU analysis of the 13 PCGs indicated that the codons encoding Leu and serine (Ser) were most abundant in *A. maculatus* (Figure 2). In the PCGs of the eight species examined, there was homology in amino acid content and codon distribution between those species (Figure 3). It was deduced that conserved amino acid sequences were found in Siluriformes [39,42,48]. Furthermore, A or T in the third position was overused compared with other synonymous codons [4]. For example, codons for leucine (TTG) and serine (TCG) were rare, whereas CTA and TCA were widespread (Figure 4).

### 3.3. Transfer RNAs and Ribosomal RNAs

Twenty-two classical structures of tRNAs were validated in *A. maculatus* mtDNA, with lengths ranging from 66 (tRNA^Cys^) to 75 bp (tRNA^Leu^), with a total length of 1558 bp (Table 1). The lowest A + T content of tRNAs was found in *A. maculatus* (55.52%), *N. thalassina* (55.52%), and *S. soldatovi* (55.44%), and the highest was found in *Mystus cavasius* (55.84%). (Table 2). On the H strand, there were fourteen tRNA genes encoded, whereas the residues are on the L strand indicated that there were four tRNA genes (Table 1). The AT (0.1283) and GC skews (−0.1631) of *A. maculatus* were similar to those of several sequenced Siluriformes mtDNAs, such as *O. platypogon* and *N. thalassina* (Table 2). The predicted tRNAs are shown in Figure 5. Except for tRNA^Ser^ (GCT), which lacks the dihydrouridine ‘DHU’ arm, all tRNAs formed typical clover-leaf secondary structures in *A. maculatus* (Figure 5). The tRNA^Ser^ ‘DHU’ arm is a large loop substitute for the conserved stem-and-loop structure. This representative characteristic [1] was also observed in the mtDNA of other Siluriformes species, including *Ompok bimaculatus* [48], *Hemibagrus* sp. [49], *Silurus lanzhouensis* [50], and *Chrysichthys nigrodigitatus* [51]. Twelve tRNA genes had at least one G-T mismatch, which caused a weak bond. In the amino acid acceptor stems of tRNA^Cys (GCA)^ and tRNA^Met (CAT)^, five T-T mismatches were observed. (Figure 5). It was also found that tRNA^Leu (TAA)^ contained an A-G mismatch. In tRNA sequences, the RNA-editing mechanism, well-known in vertebrate mtDNA, can correct unmatched base pairs [52].

According to the rRNA gene content, all rRNA genes had 52.75% A + T, which indicated a trend toward A + C, as observed in other Siluriformes [22,24]. The AT and GC skews were positive (0.2369) and negative (−0.1463), respectively (Table 2). The *A. maculatus 12S rRNA* subunit gene was 958 bp, and its *16S rRNA* subunit gene was 1675 bp, respectively. As in other fish [4], the two genes were located between tRNAPhe and tRNALeu, and separated by the tRNAVal gene (Figure 1, Table 1). The rRNA gene content of *A. maculatus* was similar to that of other Siluriformes [41].

### 3.4. The Control Region

In *A. maculatus*, the D-loop region was 1075 bp in length, which was longer than in the majority of Siluriformes and was only shorter than that in *A. seemanni*. A + T content was 62.55%, which is similar to A + T content in other Siluriformes species (Table 2). This was consistent with those of previous reports of other teleosts [45]. Additionally, both AT and GC skews were strongly negative (Table 2).

### 3.5. Overlapping and Intergenic Spacer Regions

Nine gene boundaries overlapped between adjacent genes, ranging in size from 1 to 10 bp. There was a 10 bp overlap between ATP8 and ATP6 (Table 1), which was observed in several other Siluriformes mtDNA sequences. In addition, 12 intergenic spacers, ranging in size from 1 to 31 bases, contained 66 nucleotides. *tRNA^Asn^* and *tRN^Cys^* constitute the longest intergenic spacer regions (31 bp) (Table 1), which was identified with the results in *Clarias fuscus* [41].

### 3.6. Synonymous and Nonsynonymous Substitutions

In general, Ka/Ks ratios reflect evolutionary relationships between homogenous and heterogeneous species and selective pressure at the molecular level [53,54]. A ratio of Ka/Ks > 1, Ka/Ks = 1, and Ka/Ks > 1 indicate that there has been positive selection, neutral mutation, and negative selection, respectively [55]. Four Ariidae mtDNAs (*A. maculatus*, *A. arius*, *N. thalassina*, and *O. platypogon*) were investigated for their evolutionary rate differences, and the Ka and Ks substitution rates were used to calculate sequence divergences. In all 13 PCGs of the four Ariidae, the average Ka/Ks was 0.1747 and varied from 0.015 (*COXII* between OP/AM, OP/AA, or OP/NT) to 1.672 (*ND4* between AM/NT) (Figure 6). The Ka/Ks ratios indicated that there had been strong purifying selections on multiple genes. In other words, this result showed that natural selection occurred against deleterious mutations with negative selective coefficients [56]. A high percentage of AM/NT and AA/NT variable sites was observed in *ND4* (1.672) and *ND2* (1.190) among the groups, respectively, whereas the percentage in the COXII gene were the lowest, indicating that *ND4* and *ND2* underwent positive selection and COXII was the most selectively pressured mitochondrial protein. Furthermore, compared with other species, the ratio of Ka/Ks in *A. maculatus* and *A. arius* or *N. thalassina* was the lowest in all 13 protein-coding genes, implying that these three Ariidae fish had a closer phylogenetic relationship with each other than with *O. platypogon*, which is consistent with their traditional taxonomy. There were likely differences in selection pressures between the genes, and consequently, they evolved differently. It is interesting to note that the ND2 and ND4 genes have the highest ratios, indicating strand-independent selection pressures.

### 3.7. Phylogeny

Two methods (BI and ML) were used to establish phylogenetic relationships between 21 species, including the concatenated nucleic acid (Figure 7A) and amino acid (Figure 7B) sequences of the 13 PCGs. Phylogenetic tree topologies of the two superfamilies (Ariidae, Pangasiidae, Bagridae, Mochokidae, Cl ariidae, Siluridae, and Sisoridae) were similar, and strong statistics supported the following relationship among them (Figure 7). The clustering pattern of seven superfamilies was obviously consistent with previous studies [21,22,23,24]. (According to the ML method, 15 closely related genera were identified within the seven superfamilies, and *A. maculatus* (Arius) was most closely related to *A. arius* (Arius), *N. thalassina* (Netuma), *O. platypogon* (Occidentarius), and *B. panamensis* (Bagre), which was consistent with a recent study’s findings about nucleotide sequence identity [22,23]. Netuma was most closely related to Arius. To determine the location of Ariidae within Siluriformes, further taxon sampling was required within Ariidae and related superfamilies.

## 4. Conclusions

In conclusion, the present study represents common and characteristic features of *A. maculatus* and other 26 Siluriformes mtDNA, and reveals their phylogenetic relationship. The phylogenetic results strongly support the close relationship between Arius, Netuma, Occidentarius, and Bagre. Our results will provide insight into the basics of evolutionary biology, molecular identification, and conservation of the diverse Siluriformes species, as well as the gene rearrangement process and matrilineal inheritance of *A. maculatus*.

## Figures and Tables

**Figure 1 genes-13-02128-f001:**
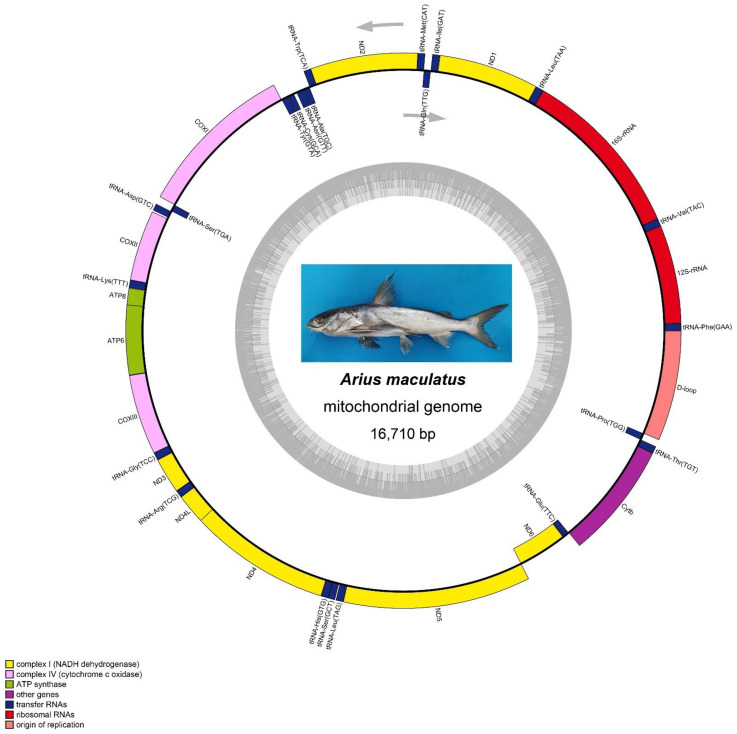
Map of the *Arius maculatus* mtDNA. Image of the *A. maculatus* was shown in the middle. The genes inside are transcribed counterclockwise, whereas the genes outside the circle are transcribed clockwise. Gene blocks are filled with different colors, as shown in the cutline. The inner ring shadow indicates the GC content of the mtDNA.

**Figure 2 genes-13-02128-f002:**
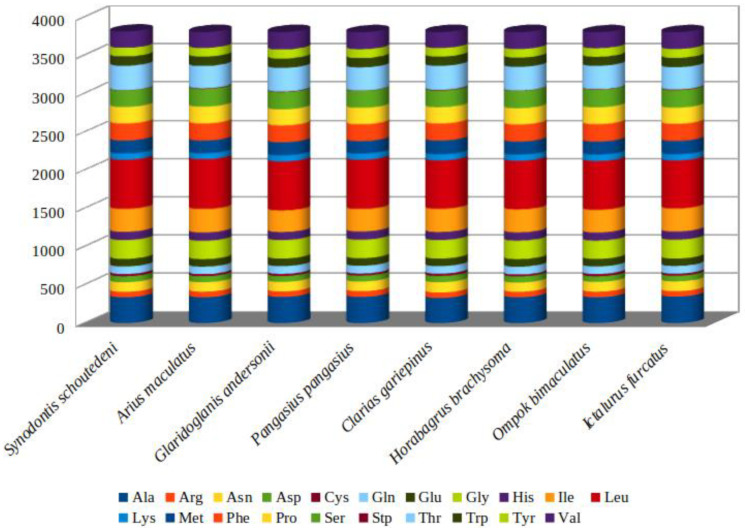
Comparison of codon usage within the mtDNA of members of the Siluriformes. Species (*A. maculatus*, Pangasius, Horabagrus brachysoma, Synodontis schoutedeni, Clarias gariepinus, Ictalurus furcatus, Ompok bimaculatus, and Glaridoglanis andersonii) represent the superfamily to which the species belongs (Ariidae, Pangasiidae, Bagridae, Mochokinae, Clariidae, Ictaluridae, Siluridae, and Sisoridae).

**Figure 3 genes-13-02128-f003:**
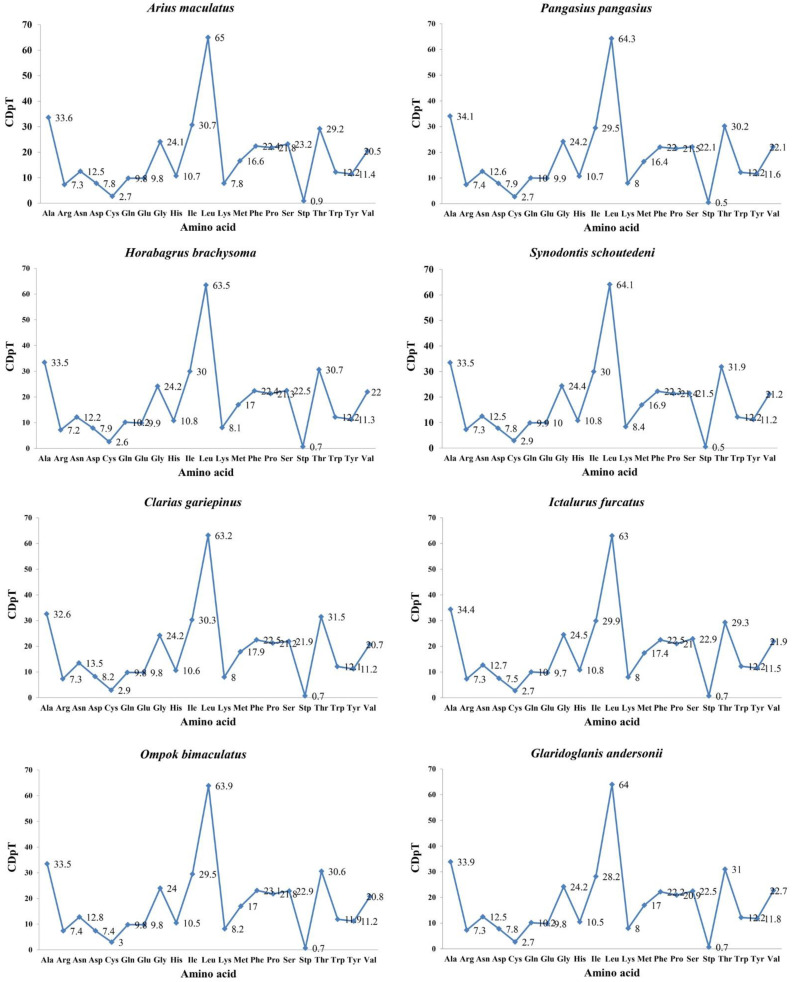
Codon distribution in members of eight superfamilies in the Siluriformes. CDspT = codons per thousand codons.

**Figure 4 genes-13-02128-f004:**
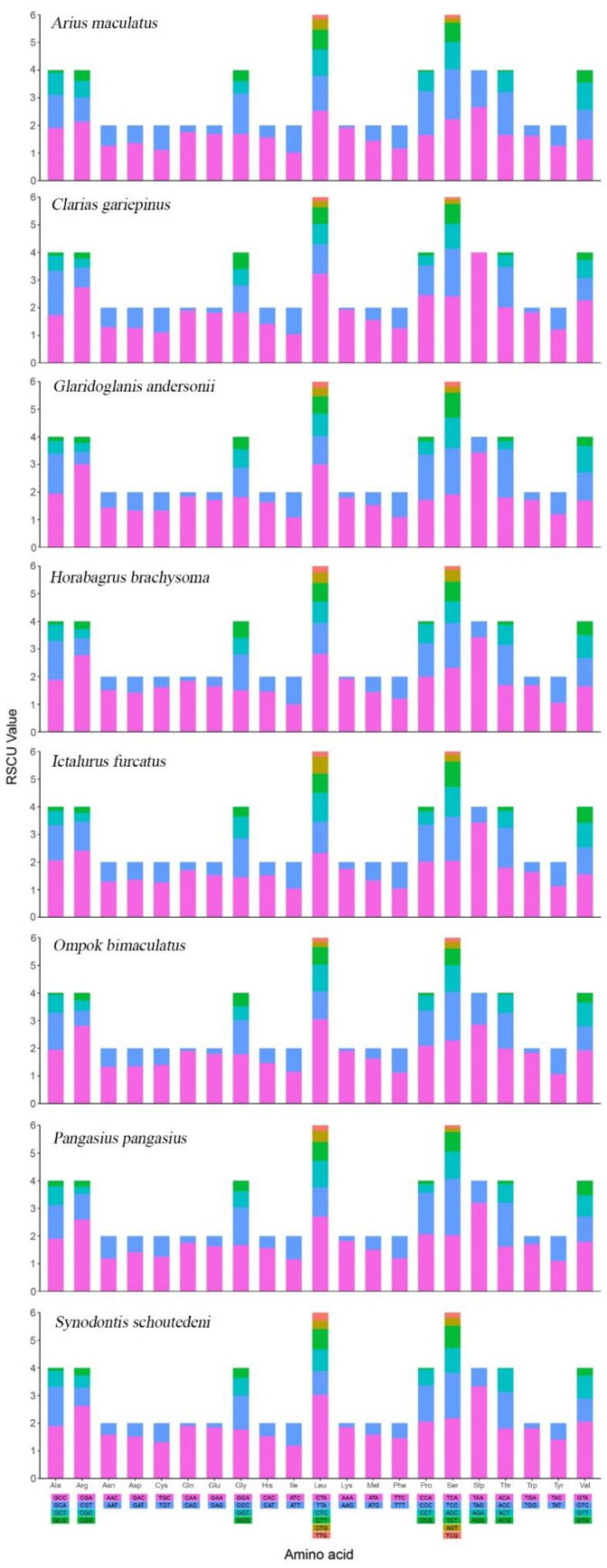
Relative synonymous codon usage (RSCU) of the mtDNA of 8 superfamilies in the Siluriformes. Codon families are plotted on the x-axis. Codons indicated above the bar are not present in the mtDNA.

**Figure 5 genes-13-02128-f005:**
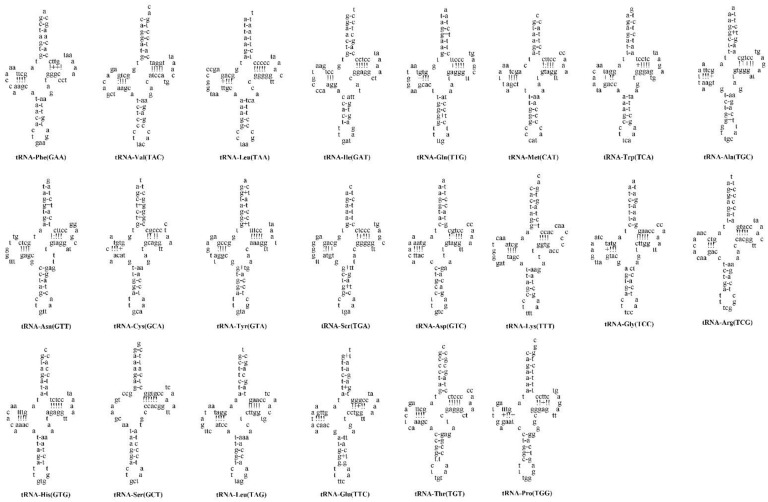
Putative secondary structures for 22 tRNA genes in mtDNA of *A. maculatus*. Watson–Crick and GT bonds are expounded by “-” and “+”, respectively.

**Figure 6 genes-13-02128-f006:**
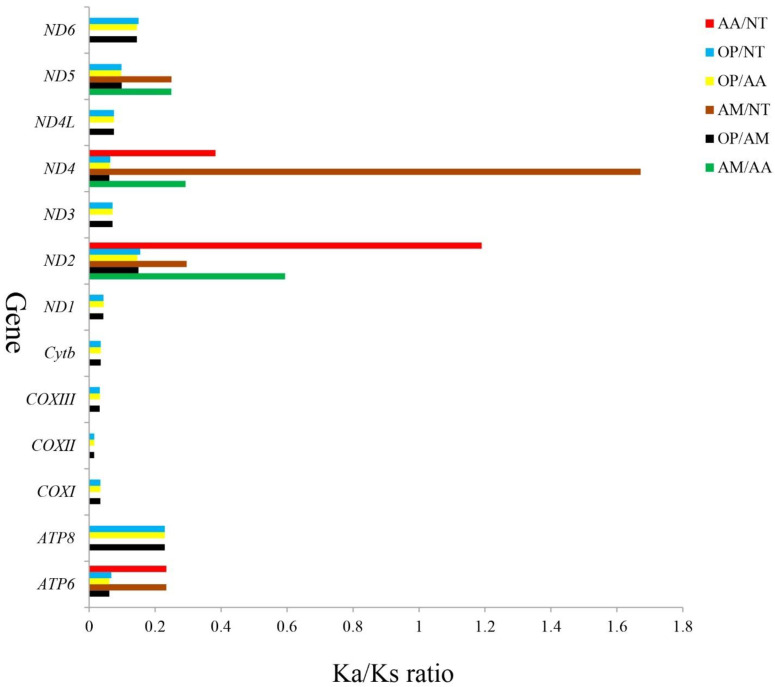
Ka/Ks ratios for the 13 PCGs among the reference *A. maculatus* (AM), *A. arius* (AA), *N. thalassina* (NT), and *O. platypogon* (OP).

**Figure 7 genes-13-02128-f007:**
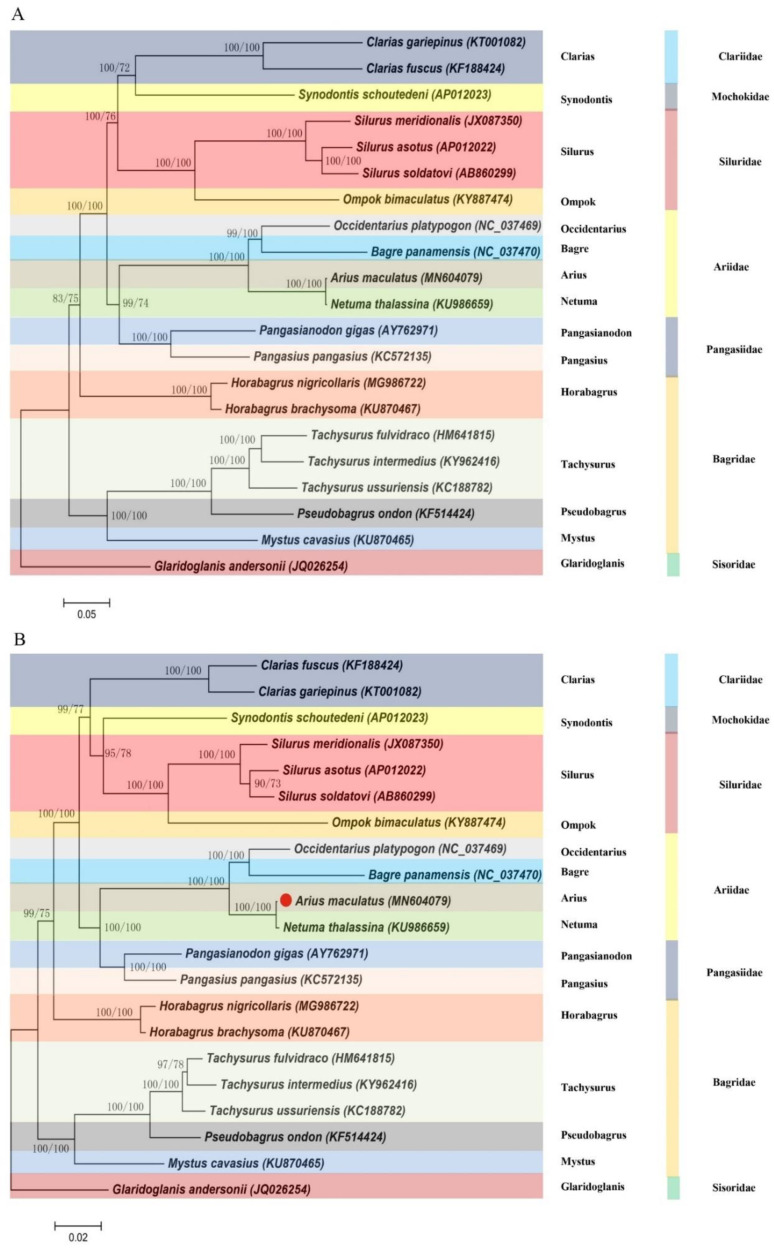
Phylogenetic trees of *A. maculatus* relationships from the (**A**) nucleotide and (**B**) amino acid datasets. Sequences alignments of mtDNA were analyzed using the RAxML and MrBayes software with the ML and BI method, respectively. Numbers at the nodes are bootstrap values (right) and Bayesian posterior probabilities (left). The accession numbers of the sequences used in the phylogenetic analysis are listed in Appendix A.

**Table 1 genes-13-02128-t001:** Sequence characteristics of *Arius maculatus* mitochondrial genome.

Locus Name	One Letter Code	From	To	Size	Strand	No. of Amino Acids	Anticodon	Inferred Initiation Codon	Inferred Termination Codon	GC_Percent	Intergenic Nucleotides
tRNA-Phe	F	1	70	70	H		GAA			40.00%	0
12S-rRNA		71	1028	958	H					49.16%	0
tRNA-Val	V	1029	1100	72	H		TAC			48.61%	0
16S-rRNA		1101	2775	1675	H					46.15%	0
tRNA-Leu	L	2776	2850	75	H		TAA			48.00%	1
ND1		2852	3826	975	H	324		ATG	TAA	47.08%	1
tRNA-Ile	I	3828	3899	72	H		GAT			50.00%	−1
tRNA-Gln	Q	3899	3969	71	L		TTG			43.66%	−1
tRNA-Met	M	3969	4038	70	H		CAT			40.00%	0
ND2		4039	5085	1047	H	348		ATG	TAG	45.75%	−2
tRNA-Trp	W	5084	5154	71	H		TCA			38.03%	2
tRNA-Ala	A	5157	5225	69	L		TGC			40.58%	1
tRNA-Asn	N	5227	5299	73	L		GTT			47.95%	31
tRNA-Cys	C	5331	5396	66	L		GCA			50.00%	1
tRNA-Tyr	Y	5398	5467	70	L		GTA			47.14%	1
COXI		5469	7019	1551	H	516		GTG	TAA	44.36%	0
tRNA-Ser	S	7020	7090	71	L		TGA			52.11%	4
tRNA-Asp	D	7095	7163	69	H		GTC			42.03%	14
COXII		7178	7868	691	H	230		ATG	T	42.11%	0
tRNA-Lys	K	7869	7942	74	H		TTT			44.59%	1
ATP8		7944	8111	168	H	55		ATG	TAA	38.69%	−10
ATP6		8102	8785	684	H	227		ATG	TAA	42.69%	−1
COXIII		8785	9568	784	H	261		ATG	T	47.19%	0
tRNA-Gly	G	9569	9641	73	H		TCC			36.99%	0
ND3		9642	9992	351	H	116		ATG	TAG	45.87%	−2
tRNA-Arg	R	9991	10,061	71	H		TCG			46.48%	0
ND4L		10,062	10,358	297	H	98		ATG	TAA	49.83%	−7
ND4		10,352	11,732	1381	H	460		ATG	T	45.84%	0
tRNA-His	H	11,733	11,802	70	H		GTG			30.00%	0
tRNA-Ser	S	11,803	11,869	67	H		GCT			50.75%	8
tRNA-Leu	L	11,878	11,950	73	H		TAG			41.10%	0
ND5		11,951	13,777	1827	H	608		ATG	TAA	43.13%	−4
ND6		13,774	14,286	513	L	170		ATG	TAG	47.76%	0
tRNA-Glu	E	14,287	14,355	69	L		TTC			36.23%	1
Cytb		14,357	15,494	1138	H	379		ATG	T	46.40%	0
tRNA-Thr	T	15,495	15,566	72	H		TGT			56.94%	−2
tRNA-Pro	P	15,565	15,634	70	L		TGG			47.14%	0
D-loop	□	15,635	16,710	1076	H	□	□	□	□	37.45%	0

+ and − correspond to the H and L strands, respectively.

**Table 2 genes-13-02128-t002:** Nucleotide composition of the mitochondrial genome in different Siluriformes.

Species	Size (bp)	A%	T%	G%	C%	A + T %	AT Skewness	GC Skewness
**Whole Mitogenome**								
*A. maculatus*	16,710	29.63	25.42	29.65	15.3	55.05	0.0764	−0.3194
*S. seemanni*	16,830	30.21	26.68	27.88	14.63	56.89	0.0619	−0.3115
*P. ondon*	16,534	31.06	25.72	27.97	15.24	56.78	0.094	−0.2946
*P. pangasius*	16,476	30.48	25.09	28.74	15.68	55.57	0.097	−0.294
*P. ussuriensis*	16,536	31.79	26.84	26.5	14.87	58.63	0.0845	−0.2811
*B. panamensis*	16,718	30.75	26.64	28.17	14.34	57.39	0.0716	−0.3253
*C. gariepinus*	16,508	32.53	24.68	27.96	14.84	57.21	0.1372	−0.3066
*I. furcatus*	16,499	29.35	25.37	29.17	16.1	54.72	0.0728	−0.2886
*G. andersonii*	16,532	31.24	24.73	28.51	15.52	55.97	0.1163	−0.2951
*O. bimaculatus*	16,482	31.67	25.13	28.3	14.9	56.8	0.1151	−0.3101
*A. occidentalis*	16,535	31.01	25.23	28.91	14.85	56.24	0.1029	−0.3212
*O. platypogon*	16,714	30.73	26.37	28.54	14.35	57.1	0.0765	−0.3308
*P. gigas*	16,533	30.42	25.5	28.42	15.66	55.92	0.0879	−0.2895
*N. thalassina*	16,711	29.65	25.38	29.69	15.28	55.03	0.0775	−0.3204
*S. soldatovi*	16,527	30.45	25.49	28.2	15.86	55.94	0.0886	−0.2799
*S. schoutedeni*	16,540	31.44	24.53	28.98	15.05	55.97	0.1235	−0.3162
*P. fulvidraco*	16,527	30.83	25.53	28.23	15.41	56.36	0.0941	−0.2937
*M. cavasius*	16,554	31.93	25.73	27.4	14.95	57.66	0.1075	−0.2942
*H. brachysoma*	16,567	31.16	25.35	28.15	15.34	56.51	0.1028	−0.2944
**Protein-Coding Genes**								
*A. maculatus*	11,407	28.78	26.10	31.33	13.79	54.88	0.0489	−0.3888
*S. seemanni*	11,403	29.59	27.84	29.44	13.13	57.43	0.0304	−0.3832
*P. ondon*	11,406	30.18	26.38	29.58	13.86	56.56	0.0671	−0.3619
*P. pangasius*	11,407	29.44	25.77	30.58	14.21	55.21	0.0664	−0.3654
*P. ussuriensis*	11,406	31.12	27.85	27.67	13.37	58.96	0.0555	−0.3484
*B. panamensis*	11,397	29.91	27.66	29.68	12.74	57.58	0.039	−0.3994
*C. gariepinus*	11,409	32.24	25.23	29.34	13.19	57.47	0.1219	−0.3796
*I. furcatus*	11,403	28.23	26.06	30.90	14.81	54.29	0.0399	−0.3519
*G. andersonii*	11,409	30.42	25.16	30.41	14.00	55.59	0.0946	−0.3696
*O. bimaculatus*	11,403	31.04	25.93	29.93	13.09	56.98	0.0897	−0.3914
*A. occidentalis*	11,405	30.04	25.76	30.86	13.34	55.8	0.0767	−0.3965
*O. platypogon*	11,403	30.01	27.19	30.13	12.66	57.2	0.0492	−0.4082
*P. gigas*	11,411	29.52	26.11	30.16	14.21	55.63	0.0614	−0.3597
*N. thalassina*	11,403	28.76	26.08	31.37	13.79	54.84	0.0488	−0.3891
*S. soldatovi*	11,409	29.29	26.62	29.63	14.46	55.91	0.0478	−0.3439
*S. schoutedeni*	11,432	30.99	24.93	30.77	13.30	55.92	0.1084	−0.3963
*P. fulvidraco*	11,406	29.86	26.09	30.00	14.05	55.95	0.0674	−0.3623
*M. cavasius*	11,406	31.18	26.18	29.20	13.45	57.36	0.0871	−0.3692
*H. brachysoma*	11,409	30.29	25.86	29.86	13.99	56.15	0.079	−0.362
**tRNA**								
*A. maculatus*	1558	31.32	24.20	25.87	18.61	55.52	0.1283	−0.1631
*S. seemanni*	1579	32.24	24.70	25.27	17.80	56.93	0.1324	−0.1735
*P. ondon*	1568	32.14	25.00	24.23	18.62	57.14	0.125	−0.131
*P. pangasius*	1564	31.71	24.87	24.42	18.99	56.59	0.1209	−0.1252
*P. ussuriensis*	1566	32.82	25.10	24.07	18.01	57.92	0.1334	−0.1442
*B. panamensis*	1561	31.96	24.33	25.80	17.91	56.29	0.1357	−0.1806
*C. gariepinus*	1560	31.60	24.94	24.68	18.78	56.54	0.1179	−0.1357
*I. furcatus*	1559	31.49	25.08	24.50	18.92	56.57	0.1134	−0.1285
*G. andersonii*	1553	31.62	24.53	24.79	19.06	56.15	0.1261	−0.1307
*O. bimaculatus*	1557	32.18	24.79	24.41	18.63	56.97	0.1297	−0.1343
*A. occidentalis*	1561	31.65	24.86	24.98	18.51	56.5	0.1202	−0.1487
*O. platypogon*	1562	31.82	24.71	25.42	18.05	56.53	0.1257	−0.1694
*P. gigas*	1559	30.98	25.02	24.70	19.31	56	0.1065	−0.1224
*N. thalassina*	1558	31.32	24.20	25.87	18.61	55.52	0.1283	−0.1631
*S. soldatovi*	1562	31.24	24.20	25.54	19.01	55.44	0.127	−0.1466
*S. schoutedeni*	1560	31.79	24.87	25.13	18.21	56.67	0.1222	−0.1598
*P. fulvidraco*	1568	32.33	24.68	24.30	18.69	57.02	0.1342	−0.1306
*M. cavasius*	1561	32.03	25.30	24.22	18.45	57.34	0.1173	−0.1351
*H. brachysoma*	1561	32.03	24.92	24.54	18.51	56.95	0.1249	−0.1399
**rRNA**								
*A. maculatus*	2633	32.62	20.13	27.08	20.17	52.75	0.2369	−0.1463
*S. seemanni*	2635	33.17	21.10	25.84	19.89	54.27	0.2224	−0.1303
*P. ondon*	2632	34.80	21.66	23.97	19.57	56.46	0.2328	−0.1012
*P. pangasius*	2633	33.38	20.74	25.48	20.40	54.12	0.2337	−0.1109
*P. ussuriensis*	2636	34.98	21.93	23.56	19.54	56.9	0.2293	−0.0933
*B. panamensis*	2636	33.69	21.48	25.36	19.47	55.17	0.2212	−0.1315
*C. gariepinus*	2627	34.64	20.25	25.43	19.68	54.89	0.2621	−0.1274
*I. furcatus*	2614	32.44	21.12	25.71	20.73	53.56	0.2114	−0.1071
*G. andersonii*	2630	33.95	20.68	25.32	20.04	54.64	0.2429	−0.1165
*O. bimaculatus*	2616	33.98	20.68	25.38	19.95	54.66	0.2434	−0.1197
*A. occidentalis*	2648	34.48	20.96	25.04	19.52	55.44	0.2439	−0.1237
*O. platypogon*	2627	33.12	20.94	26.19	19.76	54.05	0.2254	−0.14
*P. gigas*	2633	33.42	21.34	24.91	20.32	54.77	0.2205	−0.1016
*N. thalassina*	2633	32.66	20.02	27.19	20.13	52.68	0.2401	−0.1493
*S. soldatovi*	2627	34.60	20.67	24.90	19.83	55.27	0.2521	−0.1132
*S. schoutedeni*	2639	33.95	20.27	25.77	20.01	54.23	0.2523	−0.1258
*P. fulvidraco*	2631	34.70	21.63	24.02	19.65	56.33	0.2321	−0.1001
*M. cavasius*	2626	34.69	22.54	23.15	19.61	57.24	0.2122	−0.0828
*H. brachysoma*	2634	34.17	21.18	25.06	19.59	55.35	0.2346	−0.1224
**Control Region**								
*A. maculatus*	1076	29.46	33.09	23.05	14.41	62.55	−0.0579	−0.2308
*S. seemanni*	1186	27.57	30.86	20.15	12.98	58.43	−0.0563	−0.2163
*P. ondon*	891	29.97	30.98	24.92	14.14	60.94	−0.0166	−0.2759
*P. pangasius*	87	33.33	29.89	21.84	14.94	63.22	0.0545	−0.1875
*P. ussuriensis*	892	29.82	31.50	23.99	14.69	61.32	−0.0274	−0.2406
*B. panamensis*	1080	32.04	32.31	22.13	13.52	64.35	−0.0043	−0.2416
*C. gariepinus*	864	32.29	30.44	22.69	14.58	62.73	0.0295	−0.2174
*I. furcatus*	886	31.83	30.14	24.38	13.66	61.96	0.0273	−0.2819
*G. andersonii*	101	36.63	35.64	17.82	9.90	72.28	0.0137	−0.2857
*O. bimaculatus*	864	32.52	28.82	22.11	16.55	61.34	0.0604	−0.1437
*A. occidentalis*	881	32.46	31.56	22.36	13.62	64.02	0.0142	−0.2429
*O. platypogon*	1076	31.60	33.27	21.65	13.48	64.87	−0.0258	−0.2328
*P. gigas*	899	32.70	30.92	22.25	14.13	63.63	0.028	−0.2232
*N. thalassina*	1077	29.81	32.96	23.12	14.11	62.77	−0.0503	−0.2419
*S. soldatovi*	891	32.32	27.39	24.13	16.16	59.71	0.0827	−0.1978
*S. schoutedeni*	896	29.24	31.92	21.99	16.85	61.16	−0.0438	−0.1322
*P. fulvidraco*	888	29.96	31.31	24.21	14.53	61.26	−0.0221	−0.25
*M. cavasius*	910	33.85	29.78	22.09	14.29	63.63	0.0639	−0.2145
*H. brachysoma*	925	32.65	31.35	21.73	14.27	64	0.0203	−0.2072

Note: The A + T biases of whole mitogenome, protein-coding genes, tRNA, rRNA, and control regions were calculated by AT-skew= (A − T) / (A + T) and GC-skew= (G – C) / (G + C), respectively.

**Table 3 genes-13-02128-t003:** Codon usage of *A. maculatus* mitochondrial PCGs.

Amino Acid	Codon	Number	Frequency (%)	RSCU	Amino Acid	Codon	Number	Frequency (%)	RSCU
Ala	GCC	161	4.24	1.92		CAT	23	0.61	0.43
	GCA	100	2.63	1.19	Ile	ATC	156	4.10	1.02
	GCT	67	1.76	0.80		ATT	151	3.97	0.98
	GCG	8	0.21	0.10	Leu	CTA	274	7.21	2.53
Arg	CGA	39	1.03	2.14		CTC	138	3.63	1.27
	CGC	16	0.42	0.88		TTA	102	2.68	0.94
	CGG	11	0.29	0.60		CTT	78	2.05	0.72
	CGT	7	0.18	0.38		CTG	41	1.08	0.38
Asn	AAC	79	2.08	1.26		TTG	17	0.45	0.16
	AAT	46	1.21	0.74	Lys	AAA	75	1.97	1.92
Asp	GAC	53	1.39	1.36		AAG	3	0.08	0.08
	GAT	25	0.66	0.64	Met	ATA	120	3.16	1.45
Cys	TGC	15	0.39	1.11		ATG	46	1.21	0.55
	TGT	12	0.32	0.89	Phe	TTC	130	3.42	1.16
Gln	CAA	86	2.26	1.76		TTT	94	2.47	0.84
	CAG	12	0.32	0.24	Pro	CCC	90	2.37	1.65
	GAA	83	2.18	1.69		CCA	86	2.26	1.58
	GAG	15	0.39	0.31		CCT	38	1.00	0.70
Gly	GGA	102	2.68	1.69		CCG	4	0.11	0.07
	GGC	88	2.32	1.46	Ser	TCC	86	2.26	2.22
	GGG	27	0.71	0.45		TCA	70	1.84	1.81
	GGT	24	0.63	0.40		AGC	38	1.00	0.98
His	CAC	84	2.21	1.57		TCT	27	0.71	0.70
	AGT	6	0.16	0.16		ACG	4	0.11	0.05
	TCG	5	0.13	0.13	Trp	TGA	99	2.60	1.62
Stp	TAA	6	0.16	2.67		TGG	23	0.61	0.38
	TAG	3	0.08	1.33	Tyr	TAC	72	1.89	1.26
	AGA	0	0.00	0.00		TAT	42	1.10	0.74
	AGG	0	0.00	0.00	Val	GTA	77	2.03	1.50
Thr	ACC	121	3.18	1.66		GTT	55	1.45	1.07
	ACA	113	2.97	1.55		GTC	51	1.34	0.99
	ACT	54	1.42	0.74		GTG	23	0.61	0.45

## Data Availability

The generated mitochondrial DNA has been submitted to the GenBank database under the accession number MN604079. Moreover, the experimental data involved in this article can be obtained by the corresponding author.

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
