# Peer review of "Characterization of the Complete Mitochondrial Genome of the Spotted Catfish Arius maculatus (Thunberg, 1792) and Its Phylogenetic Implications"

_genes, 2022, doi:10.3390/genes13112128_

Round 1

Reviewer 1 Report

The authors have sequenced, assembled, and annotated the mitochondrial genome of Spotted sea catfish (Arius maculatus) and compared it with other mtDNA datasets. The study is fine, and the results might be helpful for evolutionary studies. However, some issues should be addressed by the authors:

1- The data Availability Statement is missing. 

2- In the conclusions section, the authors should expand the conclusions to explain the contribution of having this newly sequenced mitogenome to the scientific community and future directions.

3- I believe the results are little explored in terms of discussion. I suggest separating the results and discussion, which may be a better option. 

Author Response

The authors have sequenced, assembled, and annotated the mitochondrial genome of Spotted sea catfish (Arius maculatus) and compared it with other mtDNA datasets. The study is fine, and the results might be helpful for evolutionary studies. However, some issues should be addressed by the authors:

1- The data Availability Statement is missing.

Reply 1: Thanks for your suggestion. Now we have added it.

Data Availability Statement: The generated mitochondrial DNA has been submitted to the GenBank database under the accession number MN604079. Moreover, the experimental data involved in this article can be obtained by the corresponding author.

2- In the conclusions section, the authors should expand the conclusions to explain the contribution of having this newly sequenced mitogenome to the scientific community and future directions.

Reply 2: Thanks for your suggestion. The conclusions section was revised as: “Our results will provide insight into the basics of evolutionary biology, molecular identification, and conservation of the diverse Siluriformes species, as well as the gene rearrangement process and matrilineal inheritance of A. maculatus.”.

3- I believe the results are little explored in terms of discussion. I suggest separating the results and discussion, which may be a better option.

Reply 3: Thanks for your suggestion. Because we referred to a version that was from this journal, the results and discussion were merged together. In the present study, there was amount of mitochondrial data. We just selected the more important data and beneficial to the overall idea of the article to discuss. Moreover, the advantage of combining them together was shorten the length of the article and intuitively relate the results to the discussion. We hope to get your approval, thank you very much!

Reviewer 2 Report

The present manuscript genes-1963081 by Yang et al. describes characterization of mitochondrial genome organization in one of the species in the genus Arius. The authors were able to recover the mitogenome, conducted annotation and performed phylogenetic analyses of the major groups of Siluriformes. The results as well as the overall presentations of the manuscript are relatively clear and straightforward. However, I found that some details necessary to be added in some sections and some justifications could be better explained. For example, in the introduction section, the authors need to clearly justify why this particular species has been selected for studying among other 23 species in the genus Arius. I list my comments in more detail below and hope the authors find them useful to improve the manuscript:

1.     Introduction section

The manuscript itself is actually dealt with phylogenetic consideration of A. maculatus based on mitochondrial genome. However, I did not find any information about phylogenetic placement of this species among other species in the genus Arius from previous study. I thus recommend adding some information on this by including paper from e.g., Betancur-R, BMC Evol Biol (9) 175

2.     Introduction section, paragraph 3

Basic information for Arius maculatus was explained in this paragraph. However, it was presented in brief. I think the authors need to expand the description of this species e.g., size of the fish, morphological key features, importance of this species for human, taxonomic status etc., to help readers better understand the characteristics of this species

3.     Introduction section, paragraph 3

It seems to me that the sentences in this paragraph is somewhat disconnected. In the early sentence of the paragraph, the authors illustrate the basic information of the species. Then the authors demonstrate that order Siluriformes and the family Ariidae are fish groups with large members of species whose systematics has not been fully resolved. The authors may modify the order of the sentences. In relation to my comment no 2, the authors may divide the paragraph into two

4.     Introduction, line 69-72

The sentence is a bit confusing and I think the objectives of the study need to be clearly justified. In my opinion, the complete mitogenome of A. maculatus in this present study are not intended to be used to evaluate the taxonomy of Ariidae

5.     Materials and methods, phylogenetic analyses

The authors need to clarify in more detail about what considerations are used to incorporate 21 species to create the dataset for phylogenetic reconstruction. Is it because the availability of the mitogenome in the public database? Is it because the 21 taxa are representatives from each genus/family in the order Siluriformes?

6.     Materials and methods, phylogenetic analyses

I recommend the authors cite the previous study for 21 sequence used in this study for reference

7.     Materials and methods, phylogenetic analyses

How long the amino acid and nucleotide sequence datasets for phylogenetic reconstruction?

8.     Materials and methods, phylogenetic analyses

To the best of my knowledge, two mitogenomes from family Ariidae has been sequenced from previous work: Arius arius and Arius dispar. Are there any reasons both species were not included in the phylogenetic analyses?

9.     Materials and methods, phylogenetic analyses

In my understanding, there are two datasets for phylogenetic analysis: amino acid and nucleotide datasets. Are methods for phylogenetic analyses were conducted in the similar ways for both datasets? How about the best evolutionary models for each dataset? The authors mentioned GTR+G mode for the best evolutionary model, was this model selected by jModelTest for both datasets?

10. Materials and methods, phylogenetic analyses, line 131-132

Is this sentence error in writing? Because previous sentence has already explained that RAxML and MrBayes performed the phylogenetics analyses

11. Results and discussion

Are the results from similarity analysis done with BLAST consistent with the results from phylogenetic analysis? That closely related species have higher percentage of similarity

12. Results and discussion, line 304-305

The authors stated that phylogenetic placement of Arius maculatus is closely related with Arius arius. But I cannot find Arius arius in the Fig 7 of phylogenetic tree, instead Arius maculatus is closely related to Netuma thalassina. Can the authors clarify this point?

13. Conclusion, line 319

Which one is correct, 21 species or 26 species Siluriformes mitogenome were used in this study?

Minor:

1.     What is the accession number of the Arius maculatus mitogenome in this study?

2.     Commonly, Bayesian probability has maximum value of 1, however in the Fig 7 of the phylogenetic tree nodal support value was 100 in maximum value? Is this correct?

Author Response

Journal Title:  " Genes "

Manuscript title: Characterization of the complete mitochondrial genome of the spotted catfish Arius maculatus (Thunberg, 1792) and its phylogenetic implications 

Manuscript number: genes-1963081

Point-to-point replies to the reviewer’s comments
Dear Dr Chen,

We greatly appreciate the reviewer’s constructive, detailed and helpful comments and have done necessary changes according to the reviewer’s advice (see our reply below on detailed comments of the reviewer).

Reviewer comments: 
Reviewer #2 (Remarks to the Author): 

The present manuscript genes-1963081 by Yang et al. describes characterization of mitochondrial genome organization in one of the species in the genus Arius. The authors were able to recover the mitogenome, conducted annotation and performed phylogenetic analyses of the major groups of Siluriformes. The results as well as the overall presentations of the manuscript are relatively clear and straightforward. However, I found that some details necessary to be added in some sections and some justifications could be better explained. For example, in the introduction section, the authors need to clearly justify why this particular species has been selected for studying among other 23 species in the genus Arius. I list my comments in more detail below and hope the authors find them useful to improve the manuscript:

  1. Introduction section

The manuscript itself is actually dealt with phylogenetic consideration of A. maculatus based on mitochondrial genome. However, I did not find any information about phylogenetic placement of this species among other species in the genus Arius from previous study. I thus recommend adding some information on this by including paper from e.g., Betancur-R, BMC Evol Biol (9) 175.

Reply 1: Thanks for your suggestion. Now we have added the “Simply, the subfamilial divisions within the Ariidae (Galeichthyinae and Ariinae) were absolutely consistent among 4 reconstruction methods conducted (MP, BI, ML-RAxML, ML-Garli) and well supported (Betancur-R 2009).” in the introduction section. Moreover, “Betancur, R., R., 2009. Molecular phylogenetics and evolutionary history of ariid catfishes revisited: a comprehensive sampling. BMC Evol Biol., 9: 175. https://doi.org/10.1186/1471-2148-9-175.” was added in the Reference.

  1. Introduction section, paragraph 3

Basic information for Arius maculatus was explained in this paragraph. However, it was presented in brief. I think the authors need to expand the description of this species e.g., size of the fish, morphological key features, importance of this species for human, taxonomic status etc., to help readers better understand the characteristics of this species.

Reply 2: Thanks for your suggestion. Now “Its body long shape, broad front, lateral flat rear, general body length more than 60 cm. Furthermore, there are serrated poison glands at the base of the dorsal and pectoral spines, which cause severe pain when stabbed, and are the defensive tools of the fish. The fish has a strong smell, but it has a high fat content. Southeast coastal residents often with "angelica" and other traditional Chinese medicine to eat.” was added in the Introduction section, paragraph 3.

  1. Introduction section, paragraph 3

It seems to me that the sentences in this paragraph is somewhat disconnected. In the early sentence of the paragraph, the authors illustrate the basic information of the species. Then the authors demonstrate that order Siluriformes and the family Ariidae are fish groups with large members of species whose systematics has not been fully resolved. The authors may modify the order of the sentences. In relation to my comment no 2, the authors may divide the paragraph into two.

Reply 3: Thanks for your suggestion. We divided the paragraph into two.

  1. Introduction, line 69-72

The sentence is a bit confusing and I think the objectives of the study need to be clearly justified. In my opinion, the complete mitogenome of A. maculatus in this present study are not intended to be used to evaluate the taxonomy of Ariidae.

Reply 4: Thanks for your suggestion. Now this sentence was revised as: “Consequently, to understand the evolutionary relationships of A. maculatus in Siluri-formes and further study the population genetics in Ariidae, we sequenced the com-plete mtDNA of A. maculatus and analyzed its characteristics and evolutionary rela-tionships in present study.”.

  1. Materials and methods, phylogenetic analyses

The authors need to clarify in more detail about what considerations are used to incorporate 21 species to create the dataset for phylogenetic reconstruction. Is it because the availability of the mitogenome in the public database? Is it because the 21 taxa are representatives from each genus/family in the order Siluriformes?

Reply 5: Thanks for your suggestion. Now “The 21 mitogenome data were all download from NCBI database. Twenty-one species were divided into 15 genus, 7 family in the order Siluriformes.” was added in the Materials and methods, phylogenetic analyses.

  1. Materials and methods, phylogenetic analyses

I recommend the authors cite the previous study for 21 sequence used in this study for reference.

Reply 6: Thanks for your suggestion. In Supplementary Table 2, the information of Superfamily, Genera, Species, Size, Genbank number, and Identity of those 21 species in the Siluriformes were detailed description. Moreover, basically all of them are quoted in the text.

  1. Materials and methods, phylogenetic analyses

How long the amino acid and nucleotide sequence datasets for phylogenetic reconstruction?

Reply 7: Thanks for your suggestion. The 13 PCG nucleotide and amino acid sequences from 21 species were used to implement the phylogenetic analysis. The gene list was ATP8, COX2, COX3, CYTB, COX1, ATP6, ND4L, ND1, ND3, ND2, ND5, ND4, and ND6, and the amino acid and nucleotide sequence datasets were 3803 aa and 11440 bp, respectively.

  1. Materials and methods, phylogenetic analyses

To the best of my knowledge, two mitogenomes from family Ariidae has been sequenced from previous work: Arius arius and Arius dispar. Are there any reasons both species were not included in the phylogenetic analyses?

Reply 8: Thanks for your suggestion. Firstly, 27 twenty-seven mitogenomes from family Siluriformes (Arius maculatus, Arius arius, Netuma thalassina, Occidentarius platypogon, Ariopsis seemanni, Bagre panamensis, Pangasius pangasius, Pangasius larnaudii, Pangasianodon gigas, Pangasianodon hypophthalmus, Horabagrus brachysoma, Horabagrus nigricollaris, Tachysurus fulvidraco, Tachysurus intermedius, Tachysurus ussuriensis, Pseudobagrus ondon, Mystus cavasius, Auchenoglanis occidentalis, Synodontis schoutedeni, Clarias gariepinus, Clarias fuscus, Ictalurus furcatus, Ompok bimaculatus, Silurus soldatovi, Silurus meridionalis, Silurus asotus, and Glaridoglanis andersonii) has been chose to phylogenetic analyses. Then we took some representative species, and 21 of them were used to analysis. Moreover, Arius arius was used in this text.

  1. Materials and methods, phylogenetic analyses

In my understanding, there are two datasets for phylogenetic analysis: amino acid and nucleotide datasets. Are methods for phylogenetic analyses were conducted in the similar ways for both datasets? How about the best evolutionary models for each dataset? The authors mentioned GTR+G mode for the best evolutionary model, was this model selected by jModelTest for both datasets?

Reply 9: Thanks for your suggestion. In the present study, the alignment of a single gene was used MUSCLE v. 3.8.31 software (http://www.drive5.com/muscle/) among all species, then the genes of each species were integrated in tandem in a certain order (such as ATP8, COX2, COX3, CYTB, COX1, ATP6, ND4L, ND1, ND3, ND2, ND5, ND4, and ND6), and translated into a set of protein-coding gene sequences, subsequently the next step in the sequence analysis.

The selected sequences of DNA nucleic acid model test and amino acid model test were used jModelTest2.1.7 (https://code.google.com/p/jmodeltest2/) and Prottest3.2 (https://code.google.com/p/prottest3/), respectively. The minimum value of AIC (Akaike Information Criterion) was selected as the best model for the conformational tree. The maximum likelihood (ML) tree was implemented in RAxML 8.0.12 under the GTR-Gamma model and MtMam+I+G model for nucleic acid and amino acid tree, respectively, and node support was calculated with 1000 boot-strap replications (random seed value of 1,234,567).

  1. Materials and methods, phylogenetic analyses, line 131-132

Is this sentence error in writing? Because previous sentence has already explained that RAxML and MrBayes performed the phylogenetics analyses

Reply 10: Thanks for your suggestion. Now “The evaluation of node precision was implemented by 1000 bootstrap replicates in MEGA 6.0 with default parameters.” was deleted.

  1. Results and discussion

Are the results from similarity analysis done with BLAST consistent with the results from phylogenetic analysis? That closely related species have higher percentage of similarity.

Reply 11: Thanks for your suggestion. In Supplementary Table 2, the information of Superfamily, Genera, Species, Size, Genbank number, and Identity were shown in the Siluriformes. The results from the similarity analysis done with BLAST was basically consistent with the results from phylogenetic analysis.

  1. Results and discussion, line 304-305

The authors stated that phylogenetic placement of Arius maculatus is closely related with Arius arius. But I cannot find Arius arius in the Fig. 7 of phylogenetic tree, instead Arius maculatus is closely related to Netuma thalassina. Can the authors clarify this point?

Reply 12: Thanks for your suggestion. It was our mistake. Now we have deleted the “Arius arius (Arius)”.

  1. Conclusion, line 319

Which one is correct, 21 species or 26 species Siluriformes mitogenome were used in this study?

Reply 13: Thanks for your suggestion. A. maculatus and other 26 Siluriformes mtDNA from family Siluriformes (Arius maculatus, Arius arius, Netuma thalassina, Occidentarius platypogon, Ariopsis seemanni, Bagre panamensis, Pangasius pangasius, Pangasius larnaudii, Pangasianodon gigas, Pangasianodon hypophthalmus, Horabagrus brachysoma, Horabagrus nigricollaris, Tachysurus fulvidraco, Tachysurus intermedius, Tachysurus ussuriensis, Pseudobagrus ondon, Mystus cavasius, Auchenoglanis occidentalis, Synodontis schoutedeni, Clarias gariepinus, Clarias fuscus, Ictalurus furcatus, Ompok bimaculatus, Silurus soldatovi, Silurus meridionalis, Silurus asotus, and Glaridoglanis andersonii) were used in this study.

Minor:

  1. What is the accession number of the Arius maculatus mitogenome in this study?

Reply 1: Thanks for your suggestion. The accession number of the Arius maculatus mitogenome is MN604079.

  1. Commonly, Bayesian probability has maximum value of 1, however in the Fig 7 of the phylogenetic tree nodal support value was 100 in maximum value? Is this correct?

Reply 2: Thanks for your suggestion. We converted the phylogenetic tree nodal support value for display. It is correct.

Best regards,

Kecheng Zhu

South China Sea Fisheries Research Institute, Chinese Academy of Fishery Sciences, Guangzhou, 510300, PR China.

E-mail address: zhukecheng@scsfri.ac.cn

Round 2

Reviewer 1 Report

Thank you for addressing my concerns and comments. The current version of the manuscript has much improved and will interest the journal's readership.

Reviewer 2 Report

I have no further concerns and advice to the revised manuscript.